# Analysis of chi angle distributions in free amino acids via multiplet fitting of proton scalar couplings

Nabiha R. Syed[1], Nafisa B. Masud[1], and Colin A. Smith[1]

[1]Department of Chemistry, Wesleyan University, Middletown, CT, United States

**Correspondence:** Colin A. Smith (colin.smith@wesleyan.edu)

**Abstract.** Scalar couplings are a fundamental aspect of nuclear magnetic resonance (NMR) experiments and provide rich information about electron-mediated interactions between nuclei. $^3J$ couplings are particularly useful for determining molecular structure through the Karplus relationship, a mathematical formula used for calculating $^3J$ coupling constants from dihedral angles. In small molecules, scalar couplings are often determined through analysis of one-dimensional proton spectra. Larger proteins have typically required specialized multidimensional pulse programs designed to overcome spectral crowding and multiplet complexity. Here we present a generalized framework for fitting scalar couplings with arbitrarily complex multiplet patterns using a weak coupling model. The method is implemented in FitNMR and applicable to 1D, 2D, and 3D NMR spectra. To gain insight into the proton-proton coupling patterns present in protein side chains, we analyze a set of free amino acid 1D spectra. We show that the weak-coupling assumption is largely sufficient for fitting the majority of resonances, although there are notable exceptions. To enable structural interpretation of all couplings, we extend generalized and self-consistent Karplus equation parameterizations to all chi angles. An enhanced model of side chain motion incorporating rotamer statistics from the Protein Data Bank (PDB) is developed. Even without stereospecific assignments of the beta hydrogens, we find that two couplings are sufficient to exclude a single-rotamer model for all amino acids except proline. While most free amino acids show rotameric populations consistent with crystal structure statistics, beta-branched valine and isoleucine deviate substantially.

## 1 Introduction

The structure and dynamics of amino acid side chains is often critical for protein function. Side chains are not only an important part of the folded structure of proteins, but also key in facilitating molecular recognition, allosteric regulation, and catalysis. Nuclear magnetic resonance (NMR) is a particularly powerful technique for studying side chains as they move in solution at physiological temperatures. $^3J$ scalar couplings give the most direct information about the local structure of side chains through the mathematical relationship between the dihedral angles of rotatable bonds and $^3J$, which was originally formulated by Karplus (1963). The numerous NMR experiments for measuring protein scalar couplings have been reviewed in detail by Vuister et al. (2002). Notably, it is possible to measure every scalar coupling involved in the side chain chi 1 angle, including $^3J$(HA-HB), $^3J$(C-HB), $^3J$(N-HB), $^3J$(HA-CG), $^3J$(N-CG), and $^3J$(C-CG). (Protein Data Bank (PDB) atom names are used throughout this manuscript.)

Homonuclear proton-proton couplings, which are the focus of the present study, result in sometimes complex multiplet patterns in one-dimensional proton NMR spectra. Numerous pulse sequences have been developed to overcome this complexity and make proton-proton couplings easier to resolve and quantify in multidimensional spectra. The first was Exclusive Correlation Spectroscopy (E.COSY) (Griesinger et al., 1985, 1986, 1987), which generates cross-peak multiplets with reduced numbers of peaks and takes advantage of passive couplings to make line splitting by the active coupling visible for inspection and quantification. For $^{13}$C-labeled samples, this idea was extended using modified versions of the HCCH-COSY and HCCH-TOCSY experiments, where $^{1}J$(C-H) was used to resolve $^{3}J$(H-H). (Gemmecker and Fesik, 1991; Griesinger and Eggenberger, 1992; Emerson and Montelione, 1992) An HXYH experiment further improved experimental efficiency by simultaneously measuring both backbone and side chain $^{3}J$ couplings using $^{13}$C$^{15}$N-labeled proteins. (Tessari et al., 1995; Löhr et al., 1999) Another class of experiments, quantitative J correlation, use the ratio between a diagonal and cross peak to determine the value of the coupling constant, first demonstrated with the HNHA experiment which measures the backbone $^{3}J$(H-HA). (Vuister and Bax, 1993) An HACACB-COSY adaptation of this technique enabled quantification of side chain couplings. (Grzesiek et al., 1995)

Another approach for obtaining scalar couplings uses numerical processing of a pair of matched experimental spectra, one having in-phase peaks with the same sign and the other having anti-phase peaks with opposite signs. (Oschkinat and Freeman, 1984; Kessler et al., 1985; Titman and Keeler, 1990; Huber et al., 1993; Prasch et al., 1998) In these methods, a pair of trial anti-phase/in-phase peaks are convolved with a multiplet in the respective spectra. The coupling is determined by finding the separation between the trial peaks that results in maximum agreement between the two convolved spectra. However, peak overlap can be a problem with these types of methods because only a single multiplet is analyzed at a time.

Various approaches have been applied to directly fit peak multiplets that can handle peak overlap. SpinEvolution (Veshtort and Griffin, 2006), Quantum Mechanical Total-Line-Shape Fitting (QMTLS) in PERCH (PERCH Solutions), ChemAdder (Tiainen et al., 2014), Guided Ideographic Spin System Model Optimization (GISSMO) (Dashti et al., 2017, 2018), ANATOLIA (Cheshkov et al., 2018), and Cosmic Truth (NMR Solutions) (Achanta et al., 2021) enable fitting of a one-dimensional spectrum by iteratively optimizing parameters used to simulate the spectrum using a quantum mechanical description of the spin system(s). (Castellano and Bothner-By, 1964; Heinzer, 1977; Cheshkov and Sinitsyn, 2020) Such calculations account for cases where the chemical shift difference between two nuclei approach the value of their scalar coupling. This leads to strong coupling and the so-called "roofing effect" where peaks in the multiplet closest to the other nucleus increase in intensity and those farthest decrease in intensity. While such calculations are usually computationally intensive, methods have been developed to very rapidly simulate 1D spectra. (Castillo et al., 2011) Global Spectral Deconvolution (GSD) in Mnova NMR (Mestrelab Research) enables fitting individual peaks in 1D spectra and classification of peaks into multiplets. (Bernstein et al., 2013) More recently, deep neural networks have been combined with lineshape fitting to automatically quantify peaks in 1D spectra. (Li et al., 2023)

Several methods exist for fitting multidimensional spectra including PINT (Ahlner et al., 2013; Niklasson et al., 2017) and INFOS (Smith, 2017), but those tools do not explicitly model scalar couplings. Amplitude-Constrained Multiplet Evaluation (ACME) was developed to fit proton-proton scalar couplings in COSY cross peaks. (Delaglio et al., 2001) Explicit modeling

of scalar couplings in multidimensional spectra can also be done in Spinach (Hogben et al., 2011), which is a widely used software library optimized for simulations of large spin systems. However, like the commercially available NMRSim (Bruker) that can also simulate multidimensional spectra, the calculations can be time-consuming and are not typically used for direct spectral fitting.

Once accurate $^3J$ couplings have been measured and quantified, they can be interpreted using the Karplus relationship, which relates $^3J$ to a linear combination of $\cos\theta$ and either $\cos 2\theta$ or $\cos^2\theta$, where $\theta$ is the dihedral angle between the coupled nuclei. The three coefficients (a constant and two scaling factors for the $\cos$ functions) determine the Karplus parameterization. The coefficients for proteins have been most often determined using a large set of scalar coupling measurements for which coordinates from X-ray crystallography are also available. A structure-free approach to parameterizing scalar couplings was developed by Schmidt et al. (1999). It depends on the measurement of many different scalar couplings, each with a different relationship to the overall dihedral angle. By having many scalar couplings, both the dihedral angles of the chemical bonds and the associated Karplus parameters can be determined in a self-consistent manner. This approach was originally applied to scalar couplings in the protein backbone (Schmidt et al., 1999) and then expanded to side chains (Pérez et al., 2001). Two decades earlier, a model known as the generalized Karplus equation was parameterized primarily using data from small molecules with six membered rings. (Haasnoot et al., 1979, 1980, 1981a, b) This approach used a formula incorporating differences in electronegativity between hydrogen and the substituted heavy atoms, the orientation of the substituent relative to the hydrogen, and the electronegativities of secondary substituents.

One of the first studies into the conformational preferences of amino acids using scalar couplings examined the effect of N- and C-terminal charge states on the rotamer equilibrium. (Pachler, 1963) For individual amino acids, the ContinUous ProbabIlity Distribution (CUPID) method was developed that also incorporated information from nuclear Overhauser effect experiments. (Dzakula et al., 1992) That and numerous other methods for analyzing scalar coupling data to determine dihedral angles and distributions have been reviewed. (Kraszni et al., 2004) Scalar couplings are also used to determine conformational ensembles of proteins using the standard and generalized Karplus equations. (Steiner et al., 2012) In the context of full proteins, molecular dynamics enhanced sampling techniques have been shown to improve convergence and fit to experimental data. (Smith et al., 2021) Beyond scalar couplings, residual dipolar couplings have also been used to analyze side chain conformations in folded proteins. (Mittermaier and Kay, 2001; Li et al., 2015)

The $^3J$(H-HA) scalar coupling is dependent on the phi backbone dihedral angle and takes distinct values depending on whether the residue is part of an alpha helix or beta sheet. In most heteronuclear NMR spectra, the power output required for decoupling $^{13}$C and/or $^{15}$N in isotopically labeled proteins limits the direct-dimension acquisition time, leading to signal truncation that hinders resolution of the $^3J$(H-HA) line splitting. Increasing molecular size also broadens the linewidths, further exacerbating the resolution. However, we recently showed that through very precise modeling of signal truncation and apodization, $^3J$(H-HA) could be quantified in ordinary $^1$H$-^{15}$N 2D spectra using nonlinear least squares fitting in FitNMR. (Dudley et al., 2020) A byproduct of this fitting is that the $^1$H transverse relaxation rate, $R_2^*$, can also be quantified, which can provide valuable information about protein structure and dynamics. (Dudley et al., 2024)

**Table 1.** Isoleucine resonances table

|      | x    | x_sc                                               | 1_m0       |
|------|------|----------------------------------------------------|-----------:|
| HA   | HA   | HA-HB                                              | 859348095  |
| HB   | HB   | HA-HB HB-HG12 HB-HG13 HB-HG2 HB-HG2 HB-HG2         | 978275274  |
| HG12 | HG12 | HB-HG12 HG12-HG13 HG12-HD1 HG12-HD1 HG12-HD1       | 1099447740 |
| HG13 | HG13 | HB-HG13 HG12-HG13 HG13-HD1 HG13-HD1 HG13-HD1       | 1088697294 |
| HG2  | HG2  | HB-HG2                                             | 3413052104 |
| HD1  | HD1  | HG12-HD1 HG13-HD1                                  | 3213059843 |

Towards the ultimate goal of being able to similarly quantify side-chain $^1$H scalar couplings and $R_2^*$ values directly from multidimensional spectra of folded proteins, as well as extract accurate volumes for highly overlapped peaks, we present an analysis of the proton-proton couplings in $^1$H spectra of individual amino acids. We describe how FitNMR was enhanced to directly model complex multiplet patterns in multidimensional spectra using a simple tabular input/output format. The strengths and weaknesses of using a model that assumes purely weak-coupling interactions are illustrated. To obtain Karplus parameters, we extend a self-consistent parameterization of $^3J$(HA-HB) couplings to include $^3J$(HB-HG), $^3J$(HG-HD), and $^3J$(HD-HE). Finally, we apply an enhanced model of side chain motion incorporating prior rotameric information to determine differences in the conformational preferences between the side chains of free amino acids and those found in crystal structures.

## 2  Methods

### 2.1  Fitting couplings in multidimensional spectra

FitNMR (Dudley et al., 2020) was originally designed such that each peak in a multiplet would be a distinct entity. To allow for scalar couplings, the chemical shifts of a given peak could be made a linear combination of auxiliary chemical shift parameters, whose coefficients were chosen such that a scalar coupling in Hz could be mapped onto the ppm scale. While this functioned well for fitting simple doublets found in protein $^1$H$-^{15}$N 2D spectra, it did not scale well to other applications, especially complicated spectra with heterogeneous coupling patterns.

To address this, we developed a new way of defining NMR spectral features, for which we use the term resonances. They are defined in a comma separated values (CSV) `resonances` text file, with an example for isoleucine shown in Table 1. The first column gives the name of the resonance, which can be arbitrarily defined. FitNMR supports up to four spectral dimensions, referred to using the names x, y, z, and a, following nomenclature used by NMRPipe (Delaglio et al., 1995). The particular nucleus associated with each dimension is given in the column with the same name as the dimension. Scalar couplings active in each dimension are given in a corresponding column whose name has the _sc suffix. They are space delimited and can also be arbitrarily named, although no nucleus and scalar coupling may share the same name. A scalar coupling can appear several times to produce canonical multiplets like triplets, quartets, etc. For instance, in isoleucine the HB resonance definition

**Table 2.** Isoleucine nuclei table

|      | omega0_ppm | r2_hz |
|------|-----------:|-------|
| HA   | 3.595      | 0.565 |
| HB   | 1.908      | 0.772 |
| HG12 | 1.396      | 0.715 |
| HG13 | 1.187      | 0.696 |
| HG2  | 0.936      | 0.654 |
| HD1  | 0.864      | 0.637 |

**Table 3.** Isoleucine scalar couplings table

|           | hz     |
|-----------|--------|
| HA-HB     | 3.95   |
| HB-HG12   | 4.83   |
| HB-HG13   | 9.29   |
| HB-HG2    | 7.02   |
| HG12-HG13 | -13.49 |
| HG12-HD1  | 7.47   |
| HG13-HD1  | 7.36   |

produces a doublet of doublets of doublets of quartets, with couplings to HA, HG12, HG13 each producing a doublet and couplings to the HG2 methyl group producing a quartet. Additional columns give the volumes associated with individual spectra, referred to by FitNMR as `m0` (initial magnetization).

Each nucleus referred to in the `resonances` table is defined in the `nuclei` table, with an example for isoleucine shown in Table 2. The first column gives the nucleus name. The second `omega0_ppm` column gives the chemical shift offset, $\Omega_0$, in ppm. The third `r2_hz` column gives the transverse relaxation rate (including an inhomogeneous contribution), $R_2^*$, in Hz. The coupling table (Table 3) just has a single `hz` data column with the value of the scalar coupling in Hz. Because they are associated with saturated carbons, all $^2J$ couplings are assumed to be negative. However, in the present work the sign has no impact because all couplings are in-phase. CSV files for all fitted parameters are available in the `data/fit1d_fitnmr_output.tar.gz` file within the supplement ZIP archive.

## 2.2 Fitting amino acid 1D spectra

Starting parameters for fitting amino acid 1D NMR spectra were adapted from the Guided Ideographic Spin System Model Optimization (GISSMO) database (Dashti et al., 2017, 2018), with couplings added or removed as appropriate. Chemical shifts

were manually altered to account for differences in referencing and effects of strong coupling which FitNMR does not currently model. Standard PDB atom names were used. When two nonmethyl protons were modeled with a single chemical shift, their respective numbers were separated by a slash. For methyl protons, the last number identifying each proton was dropped from the name. Geminal proton names were assigned to follow the ordering observed in BMRB statistics (https://bmrb.io/histogram/) and do not reflect a stereospecific analysis of the fitted $^3J$ coupling values.

Fitting was done with the `refit_peaks.R` script from FitNMR 0.7. The spectra were fit in a region $\pm 0.02$ ppm from the starting peaks in each multiplet. The chemical shift was allowed to move up to 3.5 times the starting $R_2^*$ during fitting. $R_2^*$ was constrained to be 0.1 to 2 Hz and the scalar couplings were constrained to be -20 to 20 Hz.

## 2.3  Karplus parameters for side chain chi angles

When spanning a rotatable bond, $^3J$ scalar couplings provide information about the dihedral angle ($\theta$) between the two coupled atoms through the well-known Karplus relationship (Karplus, 1963):

$$^3J(\theta) = C_0 + C_1 \cos \theta + C_2 \cos 2\theta \tag{1}$$

An alternative formulation of the Karplus relationship dependent on $\cos \theta$ and $\cos^2 \theta$ terms is often used. Here we apply two enhanced forms of the Karplus equation, one that is known as the generalized Karplus equation (Haasnoot et al., 1980) and another which we will refer to as the self-consistent Karplus equation (Pérez et al., 2001). In this work both are applied to $H_1-C_1-C_2-H_2$ dihedral angles in protein side chains. The generalized Karplus equation (Haasnoot et al., 1980) is:

$$^3J(\theta) = P_1 \cos^2 \theta + P_2 \cos \theta + P_3 + \sum \Delta \chi_i^{\mathrm{g}} \left( P_4 + P_5 \cos^2(\xi_i \theta + P_6 |\Delta \chi_i^{\mathrm{g}}|) \right) \tag{2}$$

The $\Delta \chi_i^{\mathrm{g}}$ terms give the electronegativity difference between the four other substituent groups ($S_1$ to $S_4$) bonded to the central $C_1-C_2$ atom pair. They are calculated using the difference in Huggins electronegativity (Huggins, 1953) between hydrogen and the $\alpha$ atom (bonded to $C_1$ or $C_2$) and $\beta$ atoms (bonded to the $\alpha$ atom) in each substituent group:

$$\Delta \chi^{\mathrm{g}} = \Delta \chi^{\alpha} - P_7 \sum_j \Delta \chi_j^{\beta} \tag{3}$$

Here we follow the geometric standard described by Haasnoot et al. (1980) where $S_1$ and $S_2$ are bonded to $C_1$, with $S_1$ directly clockwise from $H_1$ on a Newman projection with $C_1$ in front of $C_2$, and $S_2$ directly counterclockwise from $H_1$. $S_3$ and $S_4$ are directly clockwise and counterclockwise, respectively, from $H_2$ on a Newman projection with $C_2$ in front of $C_1$. $\xi_i$ gives the sign of rotation and is +1 for $S_1/S_3$ and -1 for $S_2/S_4$.

Parameters $P_1$ to $P_7$ were derived from fits to couplings in primarily six-membered ring structures with restricted geometries. (Haasnoot et al., 1980) Parameter set B ($P_1 = 13.7$, $P_2 = -0.73$, $P_3 = 0$, $P_4 = 0.56$, $P_5 = -2.47$, $P_6 = 16.9°$, $P_7 = 0.14$) was derived from couplings with two to four substituents. Parameter set D ($P_1 = 13.22$, $P_2 = -0.99$, $P_3 = 0$, $P_4 = 0.87$, $P_5 = -2.46$, $P_6 = 19.9°$, $P_7 = 0$) was derived from couplings with three substituents. Parameter set E ($P_1 = 13.24$, $P_2 = -0.91$, $P_3 = 0$, $P_4 = 0.53$, $P_5 = -2.41$, $P_6 = 15.5°$, $P_7 = 0.19$) was derived from couplings with four substituents. Here we

follow recommendations by Haasnoot et al. (1980) using parameter sets B, D, and E for couplings with two, three, and four substituents, respectively.

For this work, we determined the complete set of parameters ($\Delta\chi_1^{\mathrm{g}}$ to $\Delta\chi_4^{\mathrm{g}}$) necessary for generalized Karplus analysis of proton-proton couplings associated with chi 1-4 by analyzing representative amino acid structures taken from the PDB Chemical Component Directory (CCD). (Westbrook et al., 2015) An example representative structure for isoleucine is shown

in Figure 1A and the parameters determined for all amino acids are given in Table A2.

The self-consistent Karplus equation (Schmidt et al., 1999) perturbs the average scalar coupling given by the $C_0$ coefficient in equation 1 using a set of increments ($\Delta C_{0,i}$) weighted by the number ($N_i$) of proton/heavy atom $\alpha$-substitutions made around the bond for a particular element type $i$:

$$^3J(\theta) = C_0 + \sum (N_i \Delta C_{0,i}) + C_1 \cos\theta + C_2 \cos 2\theta \tag{4}$$

This formulation makes it possible to extrapolate the parameterization to chemical substructures outside the training set. For side chain proton-proton $^3J$ couplings, the following previously determined (Pérez et al., 2001) coefficients and coefficient increments were used: $C_0 = 7.24$, $C_1 = -1.37$, $C_2 = 3.61$, $\Delta C_{0,\mathrm{C}} = 0.61$, $\Delta C_{0,\mathrm{O}} = -1.59$, and $\Delta C_{0,\mathrm{S}} = -1.30$ Hz. The offset for nitrogen atoms was previously defined $\Delta C_{0,\mathrm{N}} = 0$ Hz because $N_{\mathrm{N}} = 1$ for all side chain chi 1 angles, making it impossible to separate the contribution of a nitrogen substitution from the fundamental Karplus coefficient $C_0$. For the chi 1

dihedral angle, the heavy atom substitution counts were previously published. (Pérez et al., 2001) Here, we determined the number of $\alpha$-substituents for proton-proton couplings associated with chi 1-4, which are given in given in Table A3.

In equations 1, 2, and 4, the $\theta$ angle refers to the dihedral angle between the two coupled protons, which is often offset from the canonical side chain chi angle ($\chi$) by a given value, $\Delta\chi$:

$$\theta = \chi + \Delta\chi$$

In this manuscript, whenever the $\chi$ symbol has a superscript (like g, $\alpha$, or $\beta$), it refers to electronegativity. All other instances of $\chi$ refer to side chain dihedral angles. Using the CCD representative amino acid structures, we determined $\Delta\chi$ offsets (rounded to -120°, 0°, and 120°) for chi angles 1-4, which are given in given in Tables A2 and A3.

A comparison of the generalized and self-consistent Karplus equations is shown in Figure A1. Over the nine sets of unique self-consistent parameters, the range of coupling values sampled by the generalized parameters is 2.7 Hz greater on average

than the self-consistent parameters. Whereas the generalized Karplus equation was parameterized on bonds geometrically restricted by rings, the self-consistent equation was parameterized using data from a protein in solution. While efforts were made to account for the effects of protein motional averaging in the self-consistent parameterization (see below), it could be that the degree of protein motion was underestimated, resulting in less extreme Karplus curves in order to reproduce the measured scalar couplings. The generalized parameters also produce slightly higher couplings overall, with an average coupling value

0.6 Hz greater than the self-consistent parameters. Finally, averaging over the nine sets, there is a 1.5 Hz root mean square deviation between the couplings produced by the two equations.

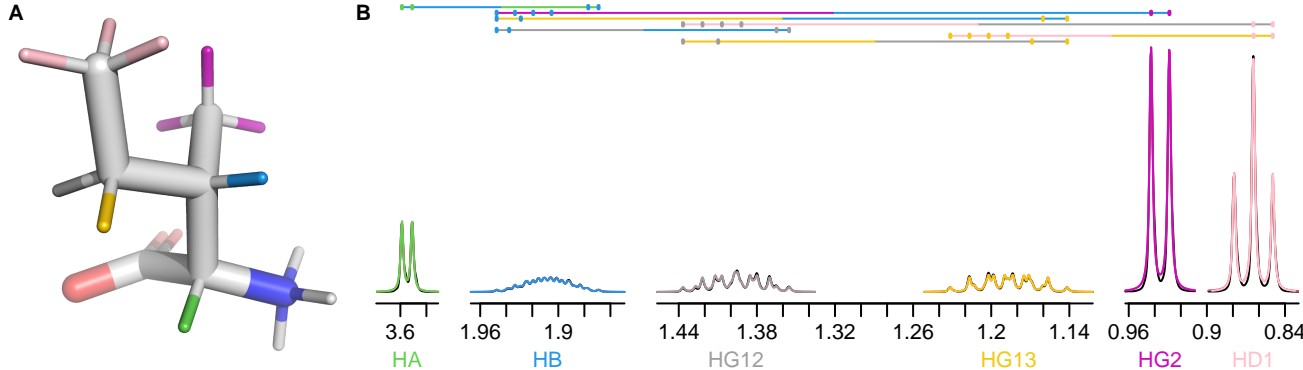

**Figure 1. A)** Representative structure of isoleucine taken from the CCD with the termini made zwitterionic in PyMOL. Protons are grouped by color, with each color having a distinct chemical shift modeled by a single resonance in the fit. The $NH_3^+$ hydrogens (white) are deuterated due to exchange with $D_2O$. **B)** Fit of 500 MHz $^1H$ NMR spectrum of isoleucine in $D_2O$ as described by Tables 1-3. The experimental spectrum is shown in black and the modeled signal corresponding to each resonance is shown with the same color as in **A**. Each scalar coupling is represented by a horizontal line above the spectrum, with the colors matched to the group being coupled to. The outermost multiplet produced by each coupling is represented by vertical dashes. The x-axis gives the $^1H$ chemical shift in ppm using a sparse representation. This panel was produced with the `plot_sparse_1d` FitNMR function.

## 2.4  Chi angle distribution analysis

During self-consistent parameterization of the side chain Karplus parameters (Pérez et al., 2001), two different models of motion were previously used. Model $M_1$ involved normally distributed fluctuation about a mean $\chi 1$ angle with standard deviation $\sigma_{\chi 1}$. Model $M_2$ assumed jumps between 60°, 180°, and 300° and varied the respective populations. Each model had two free parameters and $M_1$ was used for determining the final published parameters.

Here, we also apply a third model (which we call $M_3$) involving jumps between three rotameric bins whose chi angle distributions were taken from the 2010 Dunbrack rotamer library. (Shapovalov and Dunbrack, 2011) The $M_3$ model used 610,177 different side chain conformations from their dataset. For each side chain conformation, the theoretical coupling value was calculated using Equation 4 and the data from Table A3. Depending on the application, these theoretical couplings were averaged over all rotamers (as done for Figure 3) or the three different rotameric bins associated with chi 1 (as done for Figure 5).

## 3  Results and Discussion

### 3.1  Fitting amino acid 1D spectra

To gain insight into coupling patterns between carbon-bound protons in amino acid side chains, we performed fits of spectra taken from the Biological Magnetic Resonance Databank (BMRB) (Hoch et al., 2023). (Table A1) The samples contained

individual amino acids dissolved in $D_2O$, nearly eliminating peaks from solvent and exchangeable protons. A representative fit for isoleucine is shown in Figure 1. The resonances are defined as shown in Table 1 and parameters derived from the fit are shown in Tables 1-3. The HA, HG2, and HD1 resonances are each affected by only one or two $^3J$ couplings, making their relatively simple multiplet patterns easy to resolve. The HG12 and HG13 resonances add a mutual $^2J$ coupling and a $^3J$ coupling to the HB atom. The HG12/HG13 chemical shift difference of 104.7 Hz (relative to the -13.5 Hz $^2J$ coupling) is sufficient to minimize roofing effects from strong coupling in the experimental data (black), which shows minimal deviation from the modeled contribution of each resonance (gray and yellow, respectively). Despite a very complicated multiplet pattern for the HB atom (a doublet of doublets of doublets of quartets, blue), the resonance is very well fit by the model due to the couplings being shared with resonances having much less complexity.

Fits for all twenty amino acids are shown in Figure 2. Similar to isoleucine, the relatively simple spectra for glycine, alanine, valine, and threonine are all fit quite well and do not show significant strong coupling effects. The same is true of tryptophan, which has eight distinct resonances but only very slight strong coupling between HB2 and HB3.

Strong coupling is more pronounced in the beta protons of serine, cysteine, asparagine, and aspartate. These four side chains have the same three protons in the spectra, with HB2 and HB3 showing significant roofing effects. However, the multiplet patterns are easily resolved and the weak coupling model used by FitNMR finds an intermediate intensity between the two doublets. Despite not modeling the roofing effect, the linewidths do not appear to be distorted by the intensity mismatch. Histidine is largely similar with the addition of HD2 and HE1 nuclei in the imidazole ring that are only coupled to one another via a 1.7 Hz $^4J$ coupling.

Glutamine and glutamate add HG2 and HG3 nuclei, each with similar but distinct chemical shifts leading to large strong coupling effects. This results in outer multiplet peaks nearly disappearing. In proline, the HG2 and HG3 nuclei also have very similar chemical shifts and are quite strongly coupled. For methionine, the HG2 and HG3 nuclei appear to have indistinguishable chemical shifts and very similar scalar couplings, producing two overlapping, near-canonical triplets.

Leucine, with one HG atom, has two terminal methyl groups, each represented by a single resonance (HD1 or HD2). These make the multiplet pattern for HG quite complex. Due to the very similar $^3J$(HG-HD1) and $^3J$(HG-HD2) coupling constants (6.6 and 6.5 Hz, respectively), it is essentially a doublet of doublets of septets. Together with significant overlap between HB2, HB3, and HG (forming a strong coupling network between the three nuclei), this makes fitting the spectrum in this region very difficult. However, it is made somewhat easier because couplings involving the more isolated HA, HD1, and HD2 can be more easily resolved.

Tyrosine and phenylalanine also have somewhat complicated coupling networks in their aromatic rings, with four and five strongly coupled nuclei, respectively. They should each theoretically have both $^3J$(HD1-HE1 or HD2-HE2) and $^5J$(HD1-HE2 or HD2-HE1) couplings. In a purely weak coupling model neglecting couplings between equivalent nuclei, that would create a doublet of doublets for HD1/2 and HE1/2 in tyrosine. However, the experimental spectrum (black) resembles a doublet of triplets. The outer peaks in each triplet have much lower intensities than a classical 1:2:1 triplet and exhibit roofing. Accurate modeling of this requires separate quantum mechanical treatment of the spin states of HD1, HD2, HE1, and HE2. During fitting, $^5J$(HD-HE) drops to less than 0.0001 Hz, represented by the topmost vertical line. The phenylalanine fit does obtain

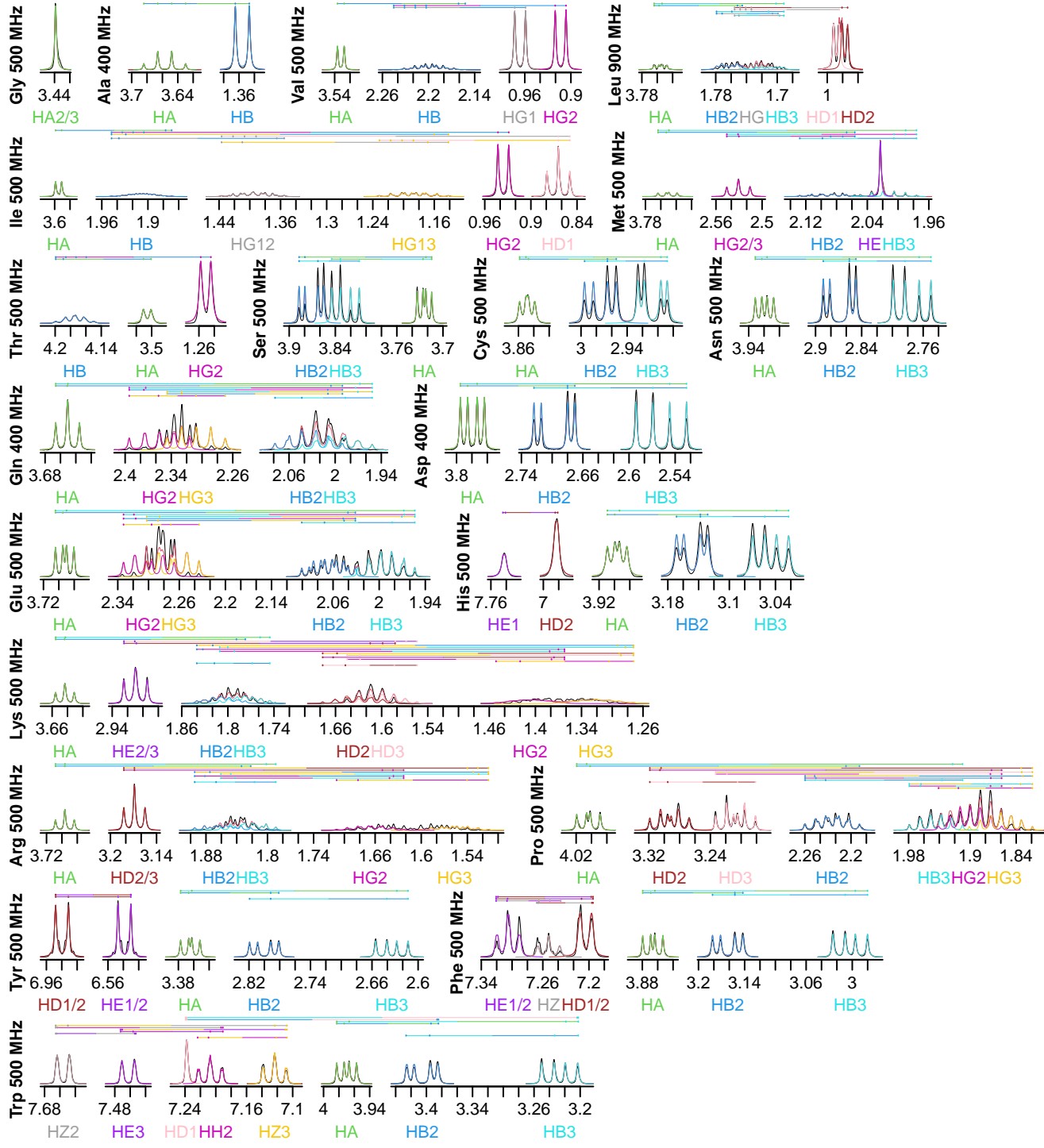

**Figure 2.** Fits of $^1$H NMR spectra of all twenty canonical amino acids in $D_2O$. Spectra are plotted as described in Figure 1. The total modeled sum of all resonance contributions is shown in red, which is usually obscured by the individual contributions.

reasonable values of 1.2 and 0.7 Hz for $^5J$(HD-HE) and $^4J$(HD-HZ), respectively. However, triplet behavior is still observed in the experimental spectrum, particularly for HE1/2, and remains unexplained by the weak coupling model.

Lysine, arginine, and proline are all capable of having distinct proton chemicals shifts at the beta, gamma, and delta positions. Distinct chemical shifts are observed for all such protons except for the HD2 and HD3 atoms in arginine. They show identical chemical shifts and produce a near perfect triplet, suggesting that the scalar couplings they make with HG2 and HG3 rotationally average out to near-identical values. The values of those scalar couplings and rotational averaging will be discussed in more detail below. While nearly all resonances in proline are modeled well, lysine and arginine are more difficult, especially for the HG2 and HG3 atoms, each of which are coupled to five nuclei.

Our data show that the large majority of protons in amino acid side chains can be modeled well using the FitNMR weak coupling approximation. However, peak overlap is an issue for several nuclei, suggesting 2D proton spectra like a NOESY or DQF-COSY may be required for adequate resolution. In addition, FitNMR and similar methods would benefit from incorporation of quantum mechanical calculations to enable accounting for strong coupling in the spectra.

## 3.2 Chi angle-dependent side chain scalar couplings

Karplus parameters are required to derive structural information from scalar couplings. For $^3J$ couplings between adjacent CH$_2$ groups, the four proton-proton couplings completely sample all three values of $\Delta\chi$ (see chi 2-4 parameters in Tables A2 and A3), providing detailed structural information. To use the generalized Karplus equation, we calculated the required electronegativity differences and positions of all substituent groups. (Table A2) For the self-consistent Karplus equation, we extrapolated parameters derived from scalar couplings associated with chi 1 (Pérez et al., 2001) to chi 2-4. (Table A3) We did not attempt a reparameterization of $C_0$ and $\Delta C_{0,N}$ (see Methods section 2.3) to account for the absence of a nitrogen substitution at chi 2 (in leucine, isoleucine, methionine, glutamine, glutamate, lysine, arginine, and proline) or chi 3 (in lysine).

The fit-derived $^3J$ couplings dependent on a side chain chi angle are shown as circles in Figure 3. There are eight amino acids with chi 2-related couplings (blue), three amino acids with chi 3-related couplings (yellow), and one with chi 4-related couplings (auburn). For chi angles with CH$_2$ groups on both sides of the associated rotatable bond, there are two couplings that in principle should take the same value due to having the same $\Delta\chi$ offset (see Methods and Tables A2/A3, and Figure A1H/I for possible exceptions). These scalar couplings were obtained without any constraint on their similarity in the software nor human knowledge of the expected equivalence during manual optimization of the input parameters. Despite that and the lack of stereospecific assignments, such equivalent couplings were within about 1 Hz of each other in all but one case (lysine HG-HD), supporting the relative accuracy of our approach despite the limitations.

As an initial point of comparison, we used the 2010 Dunbrak rotamer library dataset to calculate theoretical scalar couplings assuming the same chi angle distributions observed in crystal structures. In Figure 3, these are shown using either an x (calculated using generalized equation 2) or a plus sign (calculated with self-consistent equation 4). For the calculated rotamer library couplings, thick symbols represent the two scalar couplings with equivalent $\Delta\chi$ offsets. If our speculative stereospecific assignments for both methylene protons are either both correct or both incorrect, this pair of geometrically equivalent experimental coupling values are shown as shaded circles. Alternatively, if only one of the methylene assignments is incorrect, then

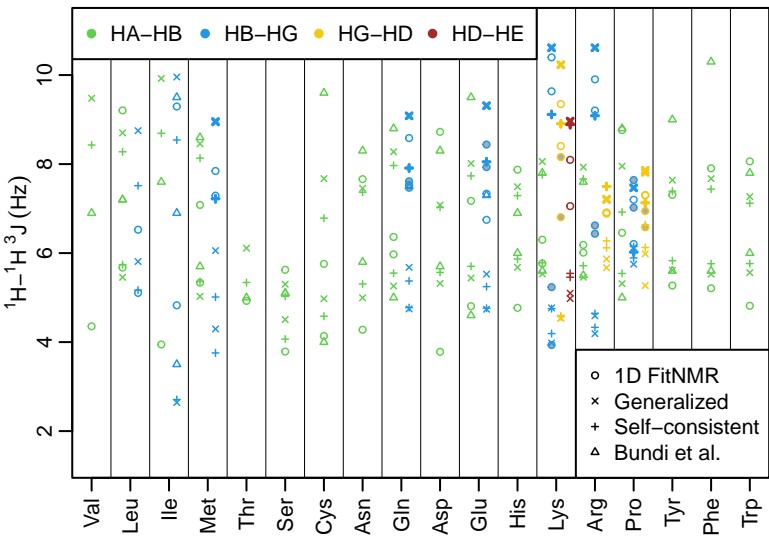

**Figure 3.** Side chain $^1H-^1H$ $^3J$ scalar couplings that depend on chi angles through the Karplus relationship. Couplings from fits of the 1D NMR spectra are shown with circles. Theoretical couplings calculated from the dataset used to create the 2010 Dunbrack rotamer library are shown with an x (using generalized Karplus equation) or a plus sign (using self-consistent equation). Experimental couplings from GGXA tetrapeptides (Bundi and Wüthrich, 1979) are shown with triangles. HA-HB couplings are shown in green, HB-HG couplings are shown in blue, HG-HD couplings are shown in yellow, and HD-HE couplings are shown in auburn. The dihedral angles governing $^3J$(HB2-HG2) and $^3J$(HB3-HG3) have the same $\Delta\chi$ (see Methods and Tables A2/A3) and therefore have the same value theoretical value shown with a thick x or plus sign. The same is true for the corresponding HG-HD and HD-HE couplings. Experimental couplings between speculatively assigned atoms 2-2 and 3-3 of adjacent methylene groups are shaded gray. Depending on the actual assignments, either the shaded or unshaded pair of experimental couplings should correspond to the theoretical couplings indicated with the thick symbols.

the unshaded circles should be equivalent. For these geometrically equivalent couplings, the experimental and rotamer library couplings are generally within about 1.5 Hz of each other. As noted above, there are possible exceptions to the coupling equivalence that happen due to geometric relationships between electron withdrawing groups and the coupled protons (Haasnoot et al., 1980), which is accounted for by the generalized Karplus equation. However, the only angle where this is observable in our simulated couplings is at proline chi 3 (see the two distinct thick yellow x symbols for proline in Figure 3 and phase shifted solid lines in Figure A1I).

Beta-branched amino acids have just a single coupling associated with chi 1, allowing unambiguous comparison. Of these, the experimental and rotamer library couplings are very similar for threonine. However, the couplings for valine and isoleucine are quite different, suggesting some combination of the charged termini, absence of neighboring amino acid residues, or solvent exposure alters the free energy of these hydrophobic residues when free in solution. For valine and isoleucine, GGXA tetrapeptide couplings (Bundi and Wüthrich, 1979) are closer to the rotamer library-derived couplings than the free amino acid couplings. For other resides, notably cysteine, glutamate, tyrosine, and phenylalanine, one of the tetrapeptide couplings is much higher than any of the free amino acid or rotamer couplings.

Many of the experimentally measured scalar couplings with ambiguous assignments have rotamer library values somewhat nearby, providing less support for (but not necessarily excluding) differences in the energetic preferences. One possible systematic divergence between the experimental and rotamer-derived couplings was in the absolute difference between the two HA-HB couplings, $\Delta^3 J(HA\text{-}HB) = |^3 J(HA\text{-}HB2) - {}^3 J(HA\text{-}HB3)|$, which is especially pronounced for aspartate. However, with the experimental $\Delta^3 J(HA\text{-}HB)$ value being greater than the rotamer library value (calculated using self-consistent equation 4) for 10 out of 15 residues, the difference was not statistically significant (p = 0.20). Likewise, using generalized Karplus equation 2 to calculate rotamer library values, only 8 out of 15 residues showed a larger experimental coupling range (p = 0.80).

### 3.3 Analysis of chi 1 angle distributions

To more quantitatively model distributions of the chi 1 angle, for which the most reliable Karplus parameters were available, we used several different models of motion. The first, $M_1$, models chi angle fluctuations as being normally distributed with standard deviations ranging from 0-50°. During development of the self-consistent Karplus parameters, both the Karplus parameters and the $M_1$ model parameters ($\chi 1$ and $\sigma_{\chi 1}$) describing each experimentally measured residue were jointly optimized to be self-consistent with one another. (Pérez et al., 2001)

For the full range of $\chi 1$ and $\sigma_{\chi 1}$ values, we calculated the root mean squared error (RMSE) between the back-calculated and experimentally measured scalar couplings. Those are shown as rectangular contour plots in Figure 4 (generalized equation 2) and Figure 5 (self-consistent equation 4). For evaluation of the different models, we made no assumption about the stereospecific assignments of the HB2 and HB3 atoms. The RMSE values were calculated for both possible assignments and the minimum RMSE for a given set of model parameters is shown. Boundaries between regions with different assignments are drawn as dashed lines. For the beta-branched amino acids (valine, isoleucine, and threonine), there is no such ambiguity but the single scalar coupling provides less information.

The chi 1 distributions used by the Dunbrack rotamer library are shown in blue on top of each contour plot. For reference, we determined the $M_1$ model parameters that produced the closest distribution (in terms of the Bhattacharyya distance) to each rotameric bin distribution. Those parameters are shown with blue plus signs. For amino acids excluding proline, the mean angles matching the rotamer library distributions (ranging 61-66°, 176-190°, 291-300°) were close to the canonical values. Due to the need for ring closure, the proline chi 1 angle distributions are skewed towards 0° or 360° and report primarily on ring pucker. The standard deviations of the rotamer library distributions ranged 6-11°, with aromatic side chains having the most variation ($\sigma_{\chi 1} \geq 10°$).

While $\sigma_{\chi 1}$ was varied 0-50° both here and in the self-consistent Karplus parameterization, $\sigma_{\chi 1}$ values much greater than those observed in the PDB are not physically realistic. Furthermore, mean angles too far from those observed in the PDB are also not likely. The applicability of the unimodal $M_1$ model to the experimental data can be judged based on how nearby a region with low RMSE is to the blue plus sign. For nearly all of the amino acids, the measured $^3 J(HA\text{-}HB)$ couplings are sufficient to exclude the $M_1$ model, suggesting that they instead populate multiple rotamer bins as would be expected for an free amino acid in solution. Proline does show a set of $M_1$ parameters with low self-consistent RMSE values very close to

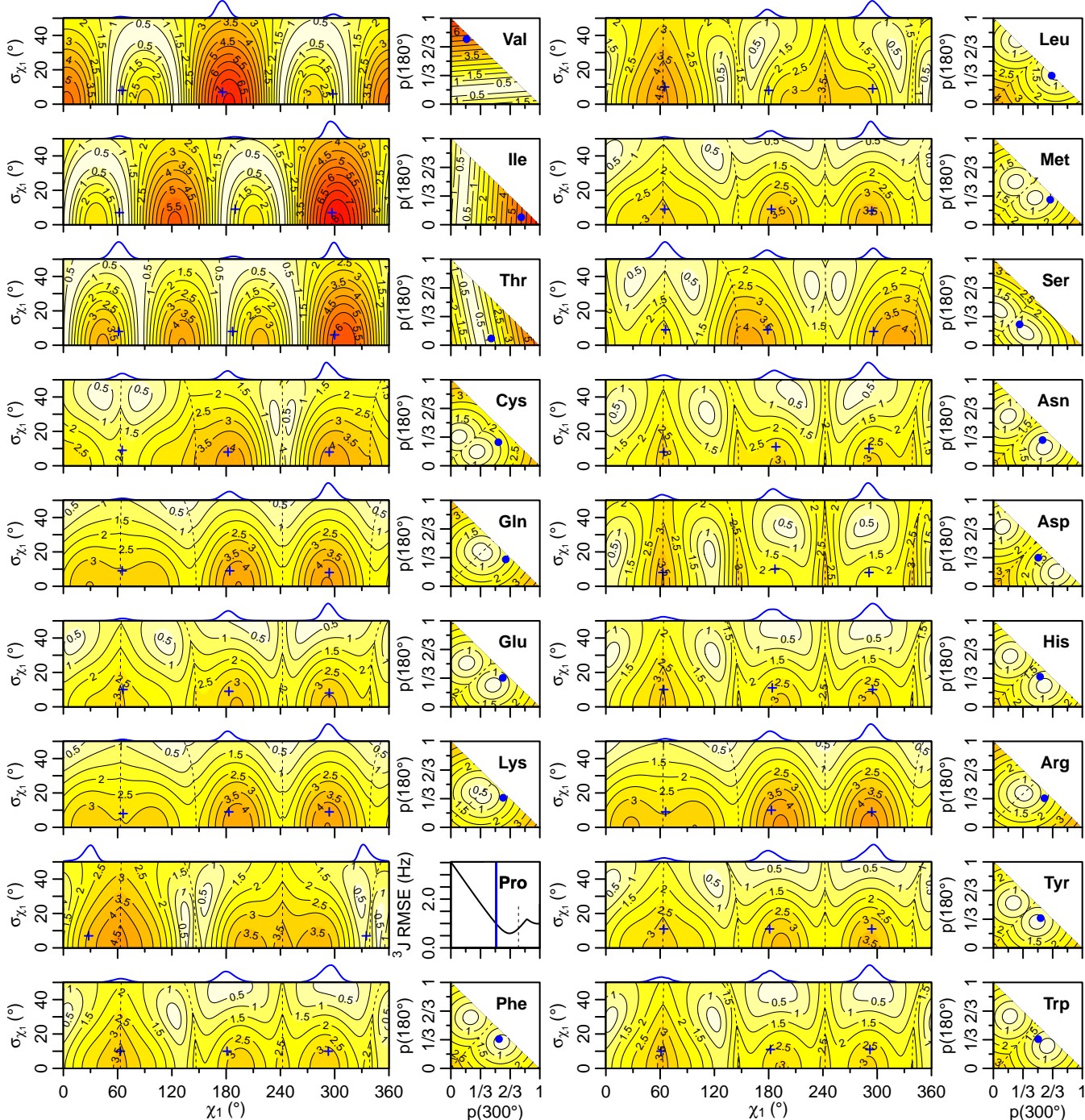

**Figure 4.** Generalized equation 2 $^3$J(HA-HB) root mean squared error (RMSE) in Hz for $M_1$ (left) and $M_3$ (right) models of motion. Dashed lines separate regions with swapped assignments. Dunbrack rotamer library distributions are shown on top of the $M_1$ plots, with the $M_1$ model having the closest match to each shown with a blue plus sign. Rotamer library populations are shown as a blue point or solid vertical line in the $M_3$ plots.

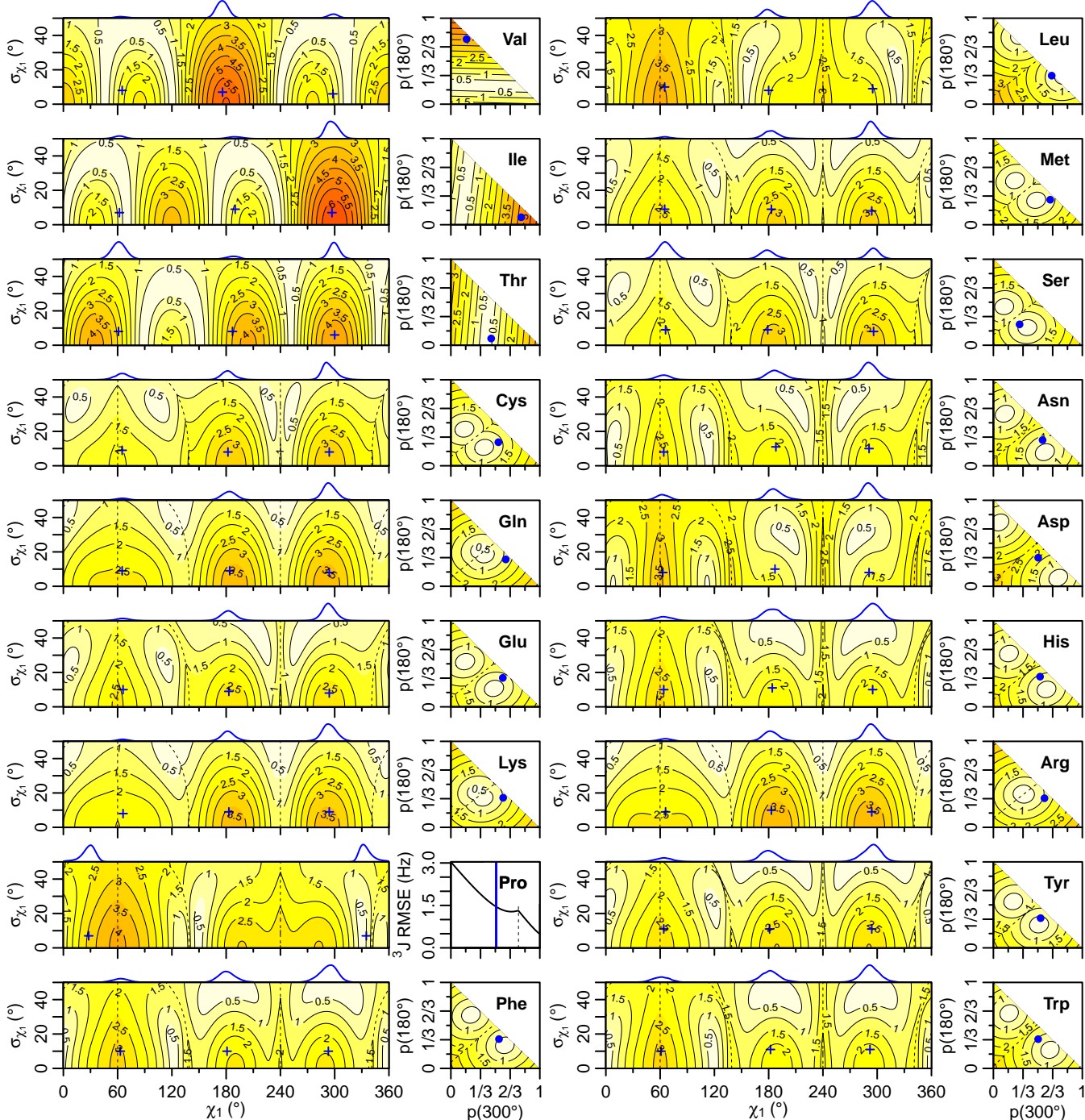

**Figure 5.** Self-consistent equation 4 $^3$J(HA-HB) root mean squared error (RMSE) in Hz for $M_1$ (left) and $M_3$ (right) models of motion. Dashed lines separate regions with swapped assignments. Dunbrack rotamer library distributions are shown on top of the $M_1$ plots, with the $M_1$ model having the closest match to each shown with a blue plus sign. Rotamer library populations are shown as a blue point or solid vertical line in the $M_3$ plots.

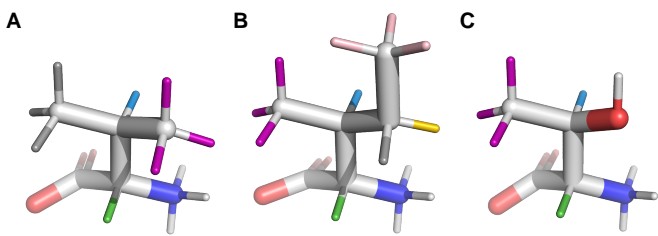

**Figure 6.** Beta-branched amino acids, with hydrogen colors matching those used in Figures 1 and 2. $^3$J(HA-HB) is most sensitive to the population of the shown rotamers because HA (green) is trans to HB (blue), giving the maximum theoretical scalar coupling. **A)** Valine $\chi1 = 180°$ rotamer. **B)** Isoleucine $\chi1 = 300°$ rotamer. **C)** Threonine $\chi1 = 300°$ rotamer. The hydroxyl hydrogen (white) was also deuterated in the sample.

a rotamer library distribution. Isoleucine is the only other amino acid where the single-rotamer model could be considered reasonable (with a plus symbol RMSE < 1 Hz), likely due to the reduced information content of the single scalar coupling.

An alternate $M_2$ model was previously tested that back-calculated the scalar couplings using a population-weighted mean of the theoretical scalar couplings at 60°, 180°, and 300°, which also makes it a two-parameter model. (Pérez et al., 2001) However, as the Dunbrack rotamer library indicates, side chains generally sample a range of values within a rotamer well. In addition, there is an amino acid-specific bias away from the canonical angles, which can be subtle for many amino acids but quite large for proline. To account for this prior information, we propose another two-parameter model, referred to here as $M_3$,

that uses average scalar couplings calculated directly from the rotameric bins in the Dunbrack 2010 rotamer library dataset.

The RMSE values for the $M_3$ model are shown as square contour plots in Figures 4 and 5, with the populations from the rotamer library shown as a blue point. Because only two valid rotamers exist for proline, the RMSE is plotted as a line vs. the population of the 300° (gauche minus) rotamer bin, with the rotamer library population shown as a vertical blue line.

For the $M_1$ model, which allows chi angles with unrealistically high potential energies, it is possible to judge model appli-

cability by comparing with rotamer library distributions (i.e. blue plus signs). However, because the $M_3$ model stays within observable chi angles by definition, there is not necessarily a means to assess model validity with *a priori* information. However, the vast majority of free amino acids do have scalar couplings reasonably consistent (RMSE < 1.5 Hz) with the rotamer library populations, which is not necessarily expected given the presence of the $NH_3^+$ and $COO^-$ groups, lack of neighboring amino acids, and high solvent exposure.

By contrast, beta-branched valine and isoleucine have $^3J$(HA-HB) values (4.4 and 3.9 Hz, respectively) quite inconsistent with the rotamer populations observed in the PDB. The $\chi1 = 180°$ rotamer of valine and $\chi1 = 300°$ rotamer of isoleucine, both highly populated in the PDB, have very similar three-dimensional structures due to differences in the way chi 1 atoms are defined. (Figure 6A/B) These rotamers are likely very prevalent in folded proteins because they avoid more strained conformations where either gamma carbon has two gauche interactions with the backbone. Interestingly, the threonine $\chi1 = 300°$

rotamer that has a similar heavy atom arrangement (Figure 6C) appears to have a 20-45% population in solution according to the $M_3$ model. (Figures 4 and 5) The differences in preferences for these rotamers in the free amino acids could arise due

to the more hydrophobic side chains of valine and isoleucine imposing a greater desolvation penalty on the $NH_3^+$ group than threonine does. The greater similarity of the GGXA tetrapeptide couplings to those from the rotamer library supports this mechanism. (Figure 3)

Proline is another amino acid whose PDB populations show varying levels of consistency with those observed for the free amino acid. Crystal structures show nearly equal populations of the $C_\gamma$ exo ($\chi 1 \approx 30°$) and $C_\gamma$ endo ($\chi 1 \approx 330°$). Interpretation of the solution NMR for free proline depends on the parameters used, with generalized equation 2 showing a 2:1 exo:endo ratio in rough agreement with the 1:1 ratio calculated by Haasnoot et al. (1981b), while self-consistent equation 4 shows a strong preference for the exo conformation. Aspartate also shows a stronger preference for either the $\chi 1 = 180°$ or $\chi 1 = 300°$ rotamers
when free in solution than it does in folded crystal structures.

Finally, several amino acids show near uniform populations of their three different chi 1 rotamers in solution, including lysine, arginine, and glutamine. All three side chains have longer aliphatic substructures, $(CH_2)_{2-4}$, and positively charged or polar head groups, which may contribute to the relatively equal rotameric free energies.

## 4   Conclusions

Our results indicate that for most nuclei, the weak-coupling assumption yields useful information about side chain dihedral angles. Only a small subset of nuclei show roofing effects from strong coupling and for nearly all that do, it results from a geminal $^2J$ coupling that does not contain readily quantifiable structural information. For the aliphatic regions of longer side chains where nuclei have both $^2J$ and $^3J$ couplings, strong coupling has a larger impact on multiplet analysis. To fully capture the complexity of multiplet patterns observed for such amino acid side chains, a strong coupling model is required. Even
in multidimensional spectra that have insufficient resolution to accurately quantify scalar couplings through computational analysis, having an accurate model of the asymmetry is likely important for quantifying the volumes of severely overlapped peaks, for instance in a 2D or 3D NOESY. As such, incorporation of a quantum mechanical spin system model into FitNMR is currently in progress.

Both the generalized (Haasnoot et al., 1980) and self-consistent (Pérez et al., 2001) Karplus equation parameterizations
appear to produce reasonable agreement between experiment and theory when extrapolated to chi 2-4, which were not part of the original training data. By mapping out the full parameter space of motion models assuming the absence ($M_1$) or presence ($M_3$) of multiple rotamers, much can be learned about side chain motion or lack thereof. As we illustrate here, differentiating between the models requires a minimum of two scalar couplings per bond. While helpful for maximum information content, stereospecific assignments do not appear strictly necessary to demonstrate the presence of multiple rotamers. While there are
multiple purely heavy atom scalar couplings associated with the chi 1 angle, the same is not true for most chi 2-4 angles. This illustrates the power of proton spectral analysis and provides motivation for further development in this area.

As shown here, peak overlap is already an issue for interpreting coupling constants from 1D spectra of individual amino acids. Overlap becomes prohibitive in 1D spectra of folded proteins but can likely be overcome to a large extent through the use of multidimensional $^1H-^1H$ 2D spectra like the NOESY, which contain at least one isolated cross-peak for many nuclei.

Without the presence of isotopically labeled heteronuclei requiring decoupling, the receiver can be left open during direct-dimension acquisition, allowing access to the complete free induction decay (FID). For small, single-digit $\mathrm{kDa}$ proteins, the multiplet patterns may be accessible to software like FitNMR in a similar manner to the $^3J$(H-HA) doublet. (Dudley et al., 2020) A relatively new class of proteins that size are computationally designed miniprotein binders, which are able to target therapeutically relevant proteins (Cao et al., 2022) and also are quite accessible to NMR characterization (Dudley et al., 2024).

Larger proteins may benefit from a strategy analogous to previously employed techniques (Oschkinat and Freeman, 1984; Kessler et al., 1985; Titman and Keeler, 1990; Huber et al., 1993; Prasch et al., 1998) of analyzing in-phase data together with anti-phase data from experiments like the DQF-COSY, where the observed signal intensity is proportional to the degree of anti-phase splitting by the coupling active in the cross-peak (Delaglio et al., 2001). Such spectra have historically been applied to the assignment and analysis of smaller unlabeled polypeptides (Wüthrich, 1986; Inagaki, 2013) but not fully exploited for

their structural information content. This study lays the groundwork for comprehensive modeling and structural interpretation of multiplets in multidimensional protein spectra.

*Code and data availability.* This manuscript was prepared using R Markdown. All code and data required for reproducing the manuscript, figures, and tables in their entirety are available in the supplement ZIP archive distributed with the paper. See the README.md file within for more details. The supplement is also available at https://github.com/smith-group/syed2024, which may be updated as necessary to maintain

software compatibility. The fitting methodology is implemented in the FitNMR open-source R package https://github.com/smith-group/fitnmr.

**Appendix A:  Supplementary information**

**A1   Processing amino acid 1D NMR data**

Amino acid 1D NMR free induction decay (FID) data were converted and processed using NMRPipe. FID conversion was

performed using the `bruker` program, with chemical shift referencing done using the temperature dependence of the $H_2O$

chemical shift. Temperatures ranged 298-306 K depending on the amino acid sample (see Table A1). Spectra were processed

with the following NMRPipe script, which includes a cosine window function and frequency domain polynomial baseline

correction:

```
#!/bin/csh
```

```
nmrPipe -in test.fid \
| nmrPipe  -fn SP -off 0.5 -end 1.00 -pow 1 -c 0.5    \
| nmrPipe  -fn ZF -auto                               \
| nmrPipe  -fn FT -auto                               \
| nmrPipe  -fn PS -p0 $P0 -p1 $P1 -di -verb           \
| nmrPipe  -fn POLY -auto                             \
   -ov -out test.ft1
```

The zero- and first-order phases were extracted from the original TopSpin processing parameters. Their signs were changed

prior to insertion into the NMRPipe script above. The Bruker PHC0 and PHC1 parameters were extracted using the following

commands:

```
grep PHC0 pdata/1/proc | cut -d " " -f 2
grep PHC1 pdata/1/proc | cut -d " " -f 2
```

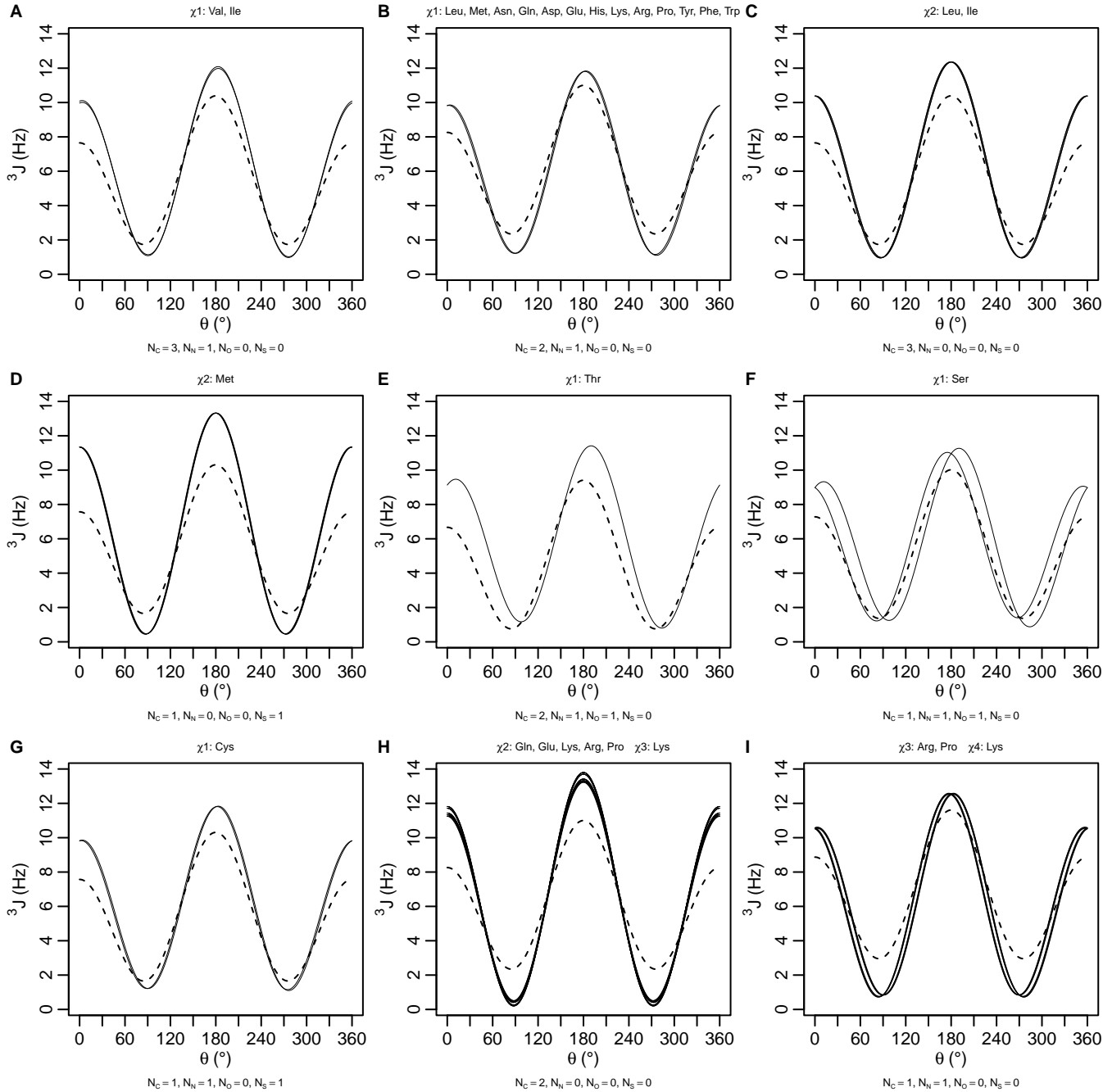

**Figure A1.** Generalized Karplus equation (thin lines) and self-consistent Karplus equation (dashed lines) for proton-proton $^3J$ scalar couplings across different chi angles. The numbers of atoms ($N_i$) used for generating the self-consistent Karplus curves are given as a subtitle. The generalized Karplus curve phase shifts observed in **F** and **I** come from large differences in the electronegativity between substituents 3 and 4 (see Table A2).

**Table A1.** Amino acid data used from the BMRB. Temperatures shown are from recorded Bruker acquisition parameters.

| Amino Acid | Field Strength (MHz) | Temperature (K) | BMRB ID |
|---|---|---|---|
| Ala | 400 | 306 | bmse000028 |
| Arg | 500 | 298 | bmse000029 |
| Asn | 500 | 298 | bmse000030 |
| Asp | 400 | 306 | bmse000031 |
| Cys | 500 | 298 | bmse000034 |
| Gln | 400 | 306 | bmse000038 |
| Glu | 500 | 298.16 | bmse000037 |
| Gly | 500 | 298 | bmse000089 |
| His | 500 | 298 | bmse000039 |
| Ile | 500 | 298 | bmse000041 |
| Leu | 900 | 298 | bmse000042 |
| Lys | 500 | 298 | bmse000043 |
| Met | 500 | 298 | bmse000044 |
| Phe | 500 | 298 | bmse000045 |
| Pro | 500 | 298 | bmse000047 |
| Ser | 500 | 298 | bmse000048 |
| Thr | 500 | 298 | bmse000049 |
| Trp | 500 | 298 | bmse000050 |
| Tyr | 500 | 298.16 | bmse000051 |
| Val | 500 | 298 | bmse000052 |

**Table A2.** Generalized Karplus equation parameters for $^1$H-$^1$H $^3$J couplings. $\Delta\chi_i^g$ give difference in substituent group electronegativity.

| AA | χ # | Δχ | H$_1$ | H$_2$ | $\Delta\chi_1^g$ | $\Delta\chi_2^g$ | $\Delta\chi_3^g$ | $\Delta\chi_4^g$ | AA | χ # | Δχ | H$_1$ | H$_2$ | $\Delta\chi_1^g$ | $\Delta\chi_2^g$ | $\Delta\chi_3^g$ | $\Delta\chi_4^g$ |
|---|---|---|---|---|---|---|---|---|---|---|---|---|---|---|---|---|---|
| Val | 1 | 0 | HA | HB | 0.850 | -0.094 | 0.400 | 0.400 | Lys | 2 | 0 | HB2 | HG2 | 0 | 0.225 | 0.344 | 0 |
| Leu | 1 | -120 | HA | HB2 | 0.850 | 0.400 | 0.400 | 0 | Lys | 2 | 120 | HB2 | HG3 | 0 | 0.225 | 0 | 0.344 |
| Leu | 1 | 0 | HA | HB3 | 0.850 | 0.400 | 0 | 0.400 | Lys | 2 | -120 | HB3 | HG2 | 0.225 | 0 | 0.344 | 0 |
| Leu | 2 | 120 | HB2 | HG | 0 | 0.400 | 0.400 | 0.400 | Lys | 2 | 0 | HB3 | HG3 | 0.225 | 0 | 0 | 0.344 |
| Leu | 2 | 0 | HB3 | HG | 0.400 | 0 | 0.400 | 0.400 | Lys | 3 | 0 | HG2 | HD2 | 0 | 0.344 | 0.281 | 0 |
| Ile | 1 | -120 | HA | HB | 0.850 | -0.094 | 0.324 | 0.400 | Lys | 3 | 120 | HG2 | HD3 | 0 | 0.344 | 0 | 0.281 |
| Ile | 2 | 0 | HB | HG12 | 0.400 | 0.400 | 0.400 | 0 | Lys | 3 | -120 | HG3 | HD2 | 0.344 | 0 | 0.281 | 0 |
| Ile | 2 | 120 | HB | HG13 | 0.400 | 0.400 | 0 | 0.400 | Lys | 3 | 0 | HG3 | HD3 | 0.344 | 0 | 0 | 0.281 |
| Met | 1 | -120 | HA | HB2 | 0.850 | 0.400 | 0.400 | 0 | Lys | 4 | 0 | HD2 | HE2 | 0 | 0.344 | 0.850 | 0 |
| Met | 1 | 0 | HA | HB3 | 0.850 | 0.400 | 0 | 0.400 | Lys | 4 | 120 | HD2 | HE3 | 0 | 0.344 | 0 | 0.850 |
| Met | 2 | 0 | HB2 | HG2 | 0 | 0.225 | 0.344 | 0 | Lys | 4 | -120 | HD3 | HE2 | 0.344 | 0 | 0.850 | 0 |
| Met | 2 | 120 | HB2 | HG3 | 0 | 0.225 | 0 | 0.344 | Lys | 4 | 0 | HD3 | HE3 | 0.344 | 0 | 0 | 0.850 |
| Met | 2 | -120 | HB3 | HG2 | 0.225 | 0 | 0.344 | 0 | Arg | 1 | -120 | HA | HB2 | 0.850 | 0.400 | 0.400 | 0 |
| Met | 2 | 0 | HB3 | HG3 | 0.225 | 0 | 0 | 0.344 | Arg | 1 | 0 | HA | HB3 | 0.850 | 0.400 | 0 | 0.400 |
| Thr | 1 | -120 | HA | HB | 0.850 | -0.094 | 1.300 | 0.400 | Arg | 2 | 0 | HB2 | HG2 | 0 | 0.225 | 0.281 | 0 |
| Ser | 1 | -120 | HA | HB2 | 0.850 | 0.400 | 1.300 | 0 | Arg | 2 | 120 | HB2 | HG3 | 0 | 0.225 | 0 | 0.281 |
| Ser | 1 | 0 | HA | HB3 | 0.850 | 0.400 | 0 | 1.300 | Arg | 2 | -120 | HB3 | HG2 | 0.225 | 0 | 0.281 | 0 |
| Cys | 1 | -120 | HA | HB2 | 0.850 | 0.400 | 0.400 | 0 | Arg | 2 | 0 | HB3 | HG3 | 0.225 | 0 | 0 | 0.281 |
| Cys | 1 | 0 | HA | HB3 | 0.850 | 0.400 | 0 | 0.400 | Arg | 3 | 0 | HG2 | HD2 | 0 | 0.344 | 0.794 | 0 |
| Asn | 1 | -120 | HA | HB2 | 0.850 | 0.400 | 0.400 | 0 | Arg | 3 | 120 | HG2 | HD3 | 0 | 0.344 | 0 | 0.794 |
| Asn | 1 | 0 | HA | HB3 | 0.850 | 0.400 | 0 | 0.400 | Arg | 3 | -120 | HG3 | HD2 | 0.344 | 0 | 0.794 | 0 |
| Gln | 1 | -120 | HA | HB2 | 0.850 | 0.400 | 0.400 | 0 | Arg | 3 | 0 | HG3 | HD3 | 0.344 | 0 | 0 | 0.794 |
| Gln | 1 | 0 | HA | HB3 | 0.850 | 0.400 | 0 | 0.400 | Pro | 1 | -120 | HA | HB2 | 0.850 | 0.400 | 0.400 | 0 |
| Gln | 2 | 0 | HB2 | HG2 | 0 | 0.225 | 0.099 | 0 | Pro | 1 | 0 | HA | HB3 | 0.850 | 0.400 | 0 | 0.400 |
| Gln | 2 | 120 | HB2 | HG3 | 0 | 0.225 | 0 | 0.099 | Pro | 2 | 0 | HB2 | HG2 | 0 | 0.225 | 0.281 | 0 |
| Gln | 2 | -120 | HB3 | HG2 | 0.225 | 0 | 0.099 | 0 | Pro | 2 | 120 | HB2 | HG3 | 0 | 0.225 | 0 | 0.281 |
| Gln | 2 | 0 | HB3 | HG3 | 0.225 | 0 | 0 | 0.099 | Pro | 2 | -120 | HB3 | HG2 | 0.225 | 0 | 0.281 | 0 |
| Asp | 1 | -120 | HA | HB2 | 0.850 | 0.400 | 0.400 | 0 | Pro | 2 | 0 | HB3 | HG3 | 0.225 | 0 | 0 | 0.281 |
| Asp | 1 | 0 | HA | HB3 | 0.850 | 0.400 | 0 | 0.400 | Pro | 3 | 0 | HG2 | HD2 | 0 | 0.344 | 0.794 | 0 |
| Glu | 1 | -120 | HA | HB2 | 0.850 | 0.400 | 0.400 | 0 | Pro | 3 | 120 | HG2 | HD3 | 0 | 0.344 | 0 | 0.794 |
| Glu | 1 | 0 | HA | HB3 | 0.850 | 0.400 | 0 | 0.400 | Pro | 3 | -120 | HG3 | HD2 | 0.344 | 0 | 0.794 | 0 |
| Glu | 2 | 0 | HB2 | HG2 | 0 | 0.225 | 0.036 | 0 | Pro | 3 | 0 | HG3 | HD3 | 0.344 | 0 | 0 | 0.794 |
| Glu | 2 | 120 | HB2 | HG3 | 0 | 0.225 | 0 | 0.036 | Tyr | 1 | -120 | HA | HB2 | 0.850 | 0.400 | 0.400 | 0 |
| Glu | 2 | -120 | HB3 | HG2 | 0.225 | 0 | 0.036 | 0 | Tyr | 1 | 0 | HA | HB3 | 0.850 | 0.400 | 0 | 0.400 |
| Glu | 2 | 0 | HB3 | HG3 | 0.225 | 0 | 0 | 0.036 | Phe | 1 | -120 | HA | HB2 | 0.850 | 0.400 | 0.400 | 0 |
| His | 1 | -120 | HA | HB2 | 0.850 | 0.400 | 0.400 | 0 | Phe | 1 | 0 | HA | HB3 | 0.850 | 0.400 | 0 | 0.400 |
| His | 1 | 0 | HA | HB3 | 0.850 | 0.400 | 0 | 0.400 | Trp | 1 | -120 | HA | HB2 | 0.850 | 0.400 | 0.400 | 0 |
| Lys | 1 | -120 | HA | HB2 | 0.850 | 0.400 | 0.400 | 0 | Trp | 1 | 0 | HA | HB3 | 0.850 | 0.400 | 0 | 0.400 |
| Lys | 1 | 0 | HA | HB3 | 0.850 | 0.400 | 0 | 0.400 | | | | | | | | | |

**Table A3.** Self-consistent Karplus equation parameters for $^1$H-$^1$H $^3$J couplings. $N_i$ give number of each type of substituent heavy atom.

| AA | $\chi$ # | $\Delta\chi$ | H$_1$ | H$_2$ | $N_C$ | $N_N$ | $N_O$ | $N_S$ | AA | $\chi$ # | $\Delta\chi$ | H$_1$ | H$_2$ | $N_C$ | $N_N$ | $N_O$ | $N_S$ |
|---|---|---|---|---|---|---|---|---|---|---|---|---|---|---|---|---|---|
| Val | 1 | 0 | HA | HB | 3 | 1 | 0 | 0 | Lys | 2 | 0 | HB2 | HG2 | 2 | 0 | 0 | 0 |
| Leu | 1 | -120 | HA | HB2 | 2 | 1 | 0 | 0 | Lys | 2 | 120 | HB2 | HG3 | 2 | 0 | 0 | 0 |
| Leu | 1 | 0 | HA | HB3 | 2 | 1 | 0 | 0 | Lys | 2 | -120 | HB3 | HG2 | 2 | 0 | 0 | 0 |
| Leu | 2 | 120 | HB2 | HG | 3 | 0 | 0 | 0 | Lys | 2 | 0 | HB3 | HG3 | 2 | 0 | 0 | 0 |
| Leu | 2 | 0 | HB3 | HG | 3 | 0 | 0 | 0 | Lys | 3 | 0 | HG2 | HD2 | 2 | 0 | 0 | 0 |
| Ile | 1 | -120 | HA | HB | 3 | 1 | 0 | 0 | Lys | 3 | 120 | HG2 | HD3 | 2 | 0 | 0 | 0 |
| Ile | 2 | 0 | HB | HG12 | 3 | 0 | 0 | 0 | Lys | 3 | -120 | HG3 | HD2 | 2 | 0 | 0 | 0 |
| Ile | 2 | 120 | HB | HG13 | 3 | 0 | 0 | 0 | Lys | 3 | 0 | HG3 | HD3 | 2 | 0 | 0 | 0 |
| Met | 1 | -120 | HA | HB2 | 2 | 1 | 0 | 0 | Lys | 4 | 0 | HD2 | HE2 | 1 | 1 | 0 | 0 |
| Met | 1 | 0 | HA | HB3 | 2 | 1 | 0 | 0 | Lys | 4 | 120 | HD2 | HE3 | 1 | 1 | 0 | 0 |
| Met | 2 | 0 | HB2 | HG2 | 1 | 0 | 0 | 1 | Lys | 4 | -120 | HD3 | HE2 | 1 | 1 | 0 | 0 |
| Met | 2 | 120 | HB2 | HG3 | 1 | 0 | 0 | 1 | Lys | 4 | 0 | HD3 | HE3 | 1 | 1 | 0 | 0 |
| Met | 2 | -120 | HB3 | HG2 | 1 | 0 | 0 | 1 | Arg | 1 | -120 | HA | HB2 | 2 | 1 | 0 | 0 |
| Met | 2 | 0 | HB3 | HG3 | 1 | 0 | 0 | 1 | Arg | 1 | 0 | HA | HB3 | 2 | 1 | 0 | 0 |
| Thr | 1 | -120 | HA | HB | 2 | 1 | 1 | 0 | Arg | 2 | 0 | HB2 | HG2 | 2 | 0 | 0 | 0 |
| Ser | 1 | -120 | HA | HB2 | 1 | 1 | 1 | 0 | Arg | 2 | 120 | HB2 | HG3 | 2 | 0 | 0 | 0 |
| Ser | 1 | 0 | HA | HB3 | 1 | 1 | 1 | 0 | Arg | 2 | -120 | HB3 | HG2 | 2 | 0 | 0 | 0 |
| Cys | 1 | -120 | HA | HB2 | 1 | 1 | 0 | 1 | Arg | 2 | 0 | HB3 | HG3 | 2 | 0 | 0 | 0 |
| Cys | 1 | 0 | HA | HB3 | 1 | 1 | 0 | 1 | Arg | 3 | 0 | HG2 | HD2 | 1 | 1 | 0 | 0 |
| Asn | 1 | -120 | HA | HB2 | 2 | 1 | 0 | 0 | Arg | 3 | 120 | HG2 | HD3 | 1 | 1 | 0 | 0 |
| Asn | 1 | 0 | HA | HB3 | 2 | 1 | 0 | 0 | Arg | 3 | -120 | HG3 | HD2 | 1 | 1 | 0 | 0 |
| Gln | 1 | -120 | HA | HB2 | 2 | 1 | 0 | 0 | Arg | 3 | 0 | HG3 | HD3 | 1 | 1 | 0 | 0 |
| Gln | 1 | 0 | HA | HB3 | 2 | 1 | 0 | 0 | Pro | 1 | -120 | HA | HB2 | 2 | 1 | 0 | 0 |
| Gln | 2 | 0 | HB2 | HG2 | 2 | 0 | 0 | 0 | Pro | 1 | 0 | HA | HB3 | 2 | 1 | 0 | 0 |
| Gln | 2 | 120 | HB2 | HG3 | 2 | 0 | 0 | 0 | Pro | 2 | 0 | HB2 | HG2 | 2 | 0 | 0 | 0 |
| Gln | 2 | -120 | HB3 | HG2 | 2 | 0 | 0 | 0 | Pro | 2 | 120 | HB2 | HG3 | 2 | 0 | 0 | 0 |
| Gln | 2 | 0 | HB3 | HG3 | 2 | 0 | 0 | 0 | Pro | 2 | -120 | HB3 | HG2 | 2 | 0 | 0 | 0 |
| Asp | 1 | -120 | HA | HB2 | 2 | 1 | 0 | 0 | Pro | 2 | 0 | HB3 | HG3 | 2 | 0 | 0 | 0 |
| Asp | 1 | 0 | HA | HB3 | 2 | 1 | 0 | 0 | Pro | 3 | 0 | HG2 | HD2 | 1 | 1 | 0 | 0 |
| Glu | 1 | -120 | HA | HB2 | 2 | 1 | 0 | 0 | Pro | 3 | 120 | HG2 | HD3 | 1 | 1 | 0 | 0 |
| Glu | 1 | 0 | HA | HB3 | 2 | 1 | 0 | 0 | Pro | 3 | -120 | HG3 | HD2 | 1 | 1 | 0 | 0 |
| Glu | 2 | 0 | HB2 | HG2 | 2 | 0 | 0 | 0 | Pro | 3 | 0 | HG3 | HD3 | 1 | 1 | 0 | 0 |
| Glu | 2 | 120 | HB2 | HG3 | 2 | 0 | 0 | 0 | Tyr | 1 | -120 | HA | HB2 | 2 | 1 | 0 | 0 |
| Glu | 2 | -120 | HB3 | HG2 | 2 | 0 | 0 | 0 | Tyr | 1 | 0 | HA | HB3 | 2 | 1 | 0 | 0 |
| Glu | 2 | 0 | HB3 | HG3 | 2 | 0 | 0 | 0 | Phe | 1 | -120 | HA | HB2 | 2 | 1 | 0 | 0 |
| His | 1 | -120 | HA | HB2 | 2 | 1 | 0 | 0 | Phe | 1 | 0 | HA | HB3 | 2 | 1 | 0 | 0 |
| His | 1 | 0 | HA | HB3 | 2 | 1 | 0 | 0 | Trp | 1 | -120 | HA | HB2 | 2 | 1 | 0 | 0 |
| Lys | 1 | -120 | HA | HB2 | 2 | 1 | 0 | 0 | Trp | 1 | 0 | HA | HB3 | 2 | 1 | 0 | 0 |
| Lys | 1 | 0 | HA | HB3 | 2 | 1 | 0 | 0 | | | | | | | | | |

*Author contributions.* NRS and NBM did fits of the 1D NMR spectra. CAS did the other analyses and wrote the manuscript.

*Competing interests.* The authors declare no competing interests.

*Disclaimer.* Publisher's note: Copernicus Publications remains neutral with regard to jurisdictional claims in published maps and institutional affiliations.

*Acknowledgements.* This work was supported by NIH grant 1R15GM141974-01.

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
