# Peer review of "Analysis of chi angle distributions in free amino acids via multiplet fitting of proton scalar couplings"

_Magnetic Resonance, 2024_

## Author Response (AR1)

**Response to Gottfried Otting**

> The article reports 3JHH couplings for the side chains of free amino acids in D2O. In the absence of a polypeptide backbone, the agreement with the predictions based on crystal structures of proteins and peptides should be best for the nuclei furthest from the backbone. Is this so?

In Figure 3, there isn't a visually discernable trend of an increase or decrease in agreement going from the HA–HB couplings to the HD–HE couplings.

> Most crystal structures are determined at cryogenic temperatures, which favour the single lowest energy conformation. Do the current data (which were recorded at which temperature?) indicate more equal populations of different rotamers?

According to the BMRB entries, the data were recorded at 298K. See for example the entry for alanine. However, the acquisition parameters recorded by the Bruker software for alanine, aspartate, and glutamine report an acquisition temperature of 306K. These temperatures are reported in Table A1.

The predicted chi1 rotameric populations can be inferred from the contour plots in Figure 4. The closer the minima are to 1/3 for p(180°) and p(300°), the more uniform the predicted populations. The blue points give the populations from PDB statistics. Based on a visual analysis, there doesn't appear to be much of a trend in the distributions based on the self-consistent equation:

- More equal populations than crystal structures: Met, Cys, Gln, Glu, Lys, Arg, Tyr
- Less equal populations than crystal structures: Leu, Ser, Asn, Asp, His, Phe, Trp

> To compare with couplings determined in peptides: how good is the agreement with the ones reported by Bundi and Wüthrich in Biopolymers 18, 285 (1979)?

We added the Budni and Wüthrich couplings to Figure 3 and make some interesting observations:

"For valine and isoleucine, GGXA tetrapeptide couplings (Bundi and Wüthrich, 1979) are closer to the rotamer library-derived couplings than the free amino acid couplings. For other resides, notably cysteine, glutamate, tyrosine, and phenylalanine, one of the tetrapeptide couplings is much higher than any of the free amino acid or rotamer couplings."

This also supports the mechanism we propose where the desolvation penalty for the N-terminal NH3+ group alters the rotameric preferences of hydrophobic beta-branched amino acids, which we now note in the text:

"The greater similarity of the GGXA tetrapeptide couplings to those from the rotamer library supports this mechanism. (Figure 3)"

> In my experience, the greatest headache with J-coupling measurements in larger proteins arises from the line broadening due to faster transverse relaxation, which collapses multiplets into unresolved fine structure. It would be very interesting to get an impression for how well the fitting algorithm can cope with significantly broader lines. Table 2 suggests that all multiplet components were narrower than 1 Hz, which is not realistic even for small proteins.

In Figure 5 of Dudley 2020, we showed through simulations that a doublet with a coalescence point of approximately 15 Hz could reliably resolved with FitNMR down to 2.5 Hz, even with a free induction decay signal to noise ratio as low as 2. As we move towards analyzing larger polypeptides in future work, where signals have coalesced, we will repeat the same types of analysis.

**Response to Anonymous Referee #1**

> Although technically the work appears fine, its level of novelty and utility to the general community appears limited in view of prior work, that the authors are probably unaware of (just a few are listed in my comments below).

We believe the primary novelty and utility of this manuscript center around the following:

- Significant enhancements made to FitNMR to allow modeling of complex multiplet structures. While this manuscript only illustrates this using 1D spectra, it works equally well with multidimensional spectra. Previously, FitNMR was not able to fit anything other than 2D HSQC-like spectra without users needing to write code. That severe limitation has now been addressed, while at the same time substantially increasing the complexity of multiplets and enabling more facile sharing of parameters between different resonances, which was critical for the present study.
- Direct comparison of protein multiplets affected by strong coupling to a model without strong coupling.
- The power and limitations of coupling constant interpretation without stereospecific assignments was explored and articulated.
- A motion model using empirical rotamer distribution data from crystal structures was proposed and applied.
- A near-complete set of coupling constants for all non-exchangeable protons in proteins was generated using PDB nomenclature for the atom names. Furthermore Karplus

parameters were determined for all 3J couplings through rotatable bonds. These should prove useful in future structural studies.

- A direct comparison between curves generated by the generalized and self-consistent Karplus equations for all protein chi angles.

> Two items that could substantially increase the value of this study:
>
> A Table with all the JHH values that were measured with their procedure.

This information was already provided in the supplement in the form of CSV files directly output from the FitNMR software. We believe that this easily parsable format is a better than a table of all these couplings, which would be quite large and cumbersome to use. In the supplement ZIP file, all of the fitted `resonances.csv`, `nuclei.csv`, and `couplings.csv` files were already provided. However, not all readers may notice the *Code and data availability* statement or understand how comprehensively the supplement captures all aspects of the work. To address this, we have included the following text in the methods:

"CSV files for all fitted parameters are available in the `data/fit1d_fitnmr_output.tar.gz` file within the supplement ZIP archive."

Furthermore, we have created a GitHub repository where the supplement files will be hosted upon publication. We plan to use this manuscript as a test of future versions of FitNMR and update the manuscript files as necessary to maintain future compatibility.

> The stereospecific assignments presumably are known from Kainosho's SAIL preparations https://doi.org/10.1017/S0033583510000016 . Not sure whether they actually published these, but I expect he will be happy to share. Alternatively, with the chemical shifts and J couplings being very similar to what they are in short peptides, it should be possible to get these assignments for at least some of the residues from a combination of NOE and JHH values, or from 3JHC and 3JHH values in the isolated amino acids

We contacted Masatsune Kainosho and SAIL Technologies and received a very kind reply. However, 1D NMR spectra for their selectively labeled amino acids are unfortunately not available. We agree that NOE and 3JHC data would be useful in making stereospecific assignments, and plan to use both types of data to make such assignments in future studies of peptides and small proteins, where the larger number of NOE contacts makes stereospecific assignment more straightforward using already published techniques.

> The analysis relies extensively on a Karplus equation with the simplified Perez substitution values, but it may be worth checking whether better agreement can be obtained with the original, very extensively researched substituent values of Haasnoot and Altona (see the 4 papers cited by Perez in their ref.36).

We were indeed not aware of this work and are very grateful to the reviewer for pointing it out.

We now make reference to this method in the introduction:

"Two decades earlier, a model known as the generalized Karplus equation was parameterized primarily using data from small molecules with six membered rings. (Haasnoot et al., 1979, 1980, 1981a, b) This approach used a formula incorporating differences in electronegativity between hydrogen and the substituted heavy atoms, the orientation of the substituent relative to the hydrogen, and the electronegativities of secondary substituents."

We carefully implemented their equation and were able to reproduce the couplings calculated in Table 3 of Haasnoot et al. 1980, although this is not shown in the manuscript. We likewise implemented code for automatically determining the required parameters from PDB files, which was similarly tested on fluorocyclohexane (in Table 3 of Haasnoot 1980).

The generalized Karplus equation analysis resulted in addition of Figure A1, Table A2, modifications to Figure 3, and Figure 4. Furthermore, extensive additions were made to Methods section 2.3. The primary things this additional analysis contributed are:

- A comparison of the Karplus curves generated using the generalized and self-consistent equations. The generalized equation parameters produce more extreme values, possibly because the self-consistent parameterization underestimated the amount of motion present in the protein used for parameter fitting.
- We show that the generalized Karplus equation produces proline rotamer populations more consistent with what is in the PDB and previously determined by Haasnoot et al. 1981. The similarity to Haasnoot's own work is expected given that they also used the generalized Karplus equation. The 1:2 population our analysis predicts is more realistic given the PDB statistics and the likely small effect the chiral C-terminal carboxylate would have on the pucker equilibrium.

Other than the proline result, we do not believe that the results shown in Figure 3 or Figures 4/5 are sufficient to make a determination of whether the generalized or self-consistent Karplus equations are better suited for analyzing protein structures. That kind of benchmarking will likely require comparison using many different NOE-based structure ensembles. That kind of study is significantly enabled by or efforts to produce the parameters given in Tables A2 and A3.

> Other than the two methods investigated by the current authors, they may also wish to consider Markley's CUPID method https://doi.org/10.1021/ja00041a044 and consider what others have found regarding sidechain rotamer populations and rotamer skewing: https://doi.org/10.1021/ja010595d and https://doi.org/10.1021/jacs.5b10072
>
> Interpretation of amino acid sidechain J HH couplings in terms of rotamer distributions spans work from over at least 6 decades, with some of the first analyses that I'm aware of by Pachler (Pachler: SpectrochimicaActa, 1963,Vol.19, pp. 2085 to 2092)

There is indeed a very significant history of analyzing rotamer distributions via scalar couplings, NOEs, residual dipolar couplings, and even relaxation dispersion experiments. We primarily tried

to introduce the reader to previously developed methodology for experimentally measuring scalar couplings, as these represent alternate approches to the direct multiplet modeling we performed. We have added the following paragraph to the manuscript to better highlight previous efforts in this area:

"One of the first studies into the conformational preferences of amino acids using scalar couplings examined the effect of N- and C-terminal charge states on the rotamer equilibrium. (Pachler, 1963) For individual amino acids, the ContinUous ProbabIlity Distribution (CUPID) method was developed that also incorporated information from nuclear Overhauser effect experiments. (Dzakula et al., 1992) That and numerous other methods for analyzing scalar coupling data to determine dihedral angles and distributions have been reviewed. (Kraszni et al., 2004) Scalar couplings are also used to determine conformational ensembles of proteins using the standard and generalized Karplus equations. (Steiner et al., 2012) In the context of full proteins, molecular dynamics enhanced sampling techniques have been shown to improve convergence and fit to experimental data. (Smith et al., 2021) Beyond scalar couplings, residual dipolar couplings have also been used to analyze side chain conformations in folded proteins. (Mittermaier and Kay, 2001; Li et al., 2015)"

> Minor Issues:
>
> 1. Table 3: clarify how you identified the negative sign of J(HG12–HG13)

We inserted text to clarify this:

"Because they are associated with saturated carbons, all 2J couplings are assumed to be negative. However, in the present work the sign has no impact because all couplings are in-phase."

> 2. Include the fitted J values as an SI table

This was addressed as described above.

> 3. Line 191: clarify what you mean with "arises due to the strong coupling network". Isn't this simply due to the 4-bond HD1–HD2 coupling for cases where HE1 and HE2 have different spin states, and similarly for HE1–HE2 J coupling for cases where HD1 and HD2 have different spin states. 5J(HE–HD) would simply be in the weak coupling limit and yield a small symmetric splitting of each of the doublet components that appears unresolved in the spectra shown.

We have corrected this section of the text to give a more accurate explanation for the observed splitting pattern:

"In a purely weak coupling model neglecting couplings between equivalent nuclei, that would

create a doublet of doublets for HD1/2 and HE1/2 in tyrosine. However, the experimental spectrum (black) resembles a doublet of triplets. The outer peaks in each triplet have much lower intensities than a classical 1:2:1 triplet and exhibit roofing. Accurate modeling of this requires separate quantum mechanical treatment of the spin states of HD1, HD2, HE1, and HE2."

> 4. The legend to Fig. 3 is a bit confusing. It says: "The dihedral angles governing 3J(HB2–HG2) and 3J(HB3–HG3) have the same Δχ" but since you don't have stereo assignments, how do you deal with this. It's also strange to see the bold HB-HG J values to be so different for Lys and Arg. See also line 230. Something must be amiss in the interpretation of the data when concluding a gauche chi2 angle for the free amino acid. Steric clashing is expected to lower this population.

We thank the reviewer for noting this oversight. Due to the lack of stereospecific assignments, we are unable to uniquely say which are the HB2-HG2 and HB3-HG3 couplings. However, we can say that one of two pairs of fitted couplings must correspond to these geometrically equivalent couplings. We have updated the figure and text to reflect this:

- Figure 3 legend: "Experimental couplings between speculatively assigned atoms 2-2 and 3-3 of adjacent methylene groups are shaded gray. Depending on the actual assignments, either the shaded or unshaded pair of experimental couplings should correspond to the theoretical couplings indicated with the thick plus symbol."

- Text: "If our speculative stereospecific assignments for both methylene protons are either both correct or both incorrect, this pair of geometrically equivalent experimental coupling values are shown as shaded circles. Alternatively, if only one of the methylene assignments is incorrect, then the unshaded circles should be equivalent."

> 5. Can the large difference between J(HA-HB2) and J(HA-HB3) be used for stereo assignment of the two resonances?

These large differences could be used for stereospecific assignment if one assumes that the 300° rotameric bin is more populated than the 180° rotameric bin for all residues. However, PDB statistics (Figure 4) show that this population bias is not very strong for non-beta branched amino acids. The only residue for which stereospecific assignments are possible would be proline, where the ring structure forbids the 180° rotamer.

> 6. Line 213-215: Perhaps try the Haasnoot values?

This was addressed above.

> 7. Line 284: list the values you found for 3J(HA-HB) for Val and Ile when discussing the issue

We added the coupling values:

"By contrast, beta-branched valine and isoleucine have 3J(HA-HB) values (4.4 and 3.9 Hz, respectively) quite inconsistent with the rotamer populations observed in the PDB."

> 8. Line 302: "where such 2J couplings are also involved in 3J couplings" Poor phrasing

We have rephrased this sentence as follows:

"For the aliphatic regions of longer side chains where nuclei have both 2J and 3J couplings, strong coupling has a larger impact on multiplet analysis."

> 9. Line 309-311: this proposes to do what Altona&Haasnoot did successfully some 40 years ago

These sentences were deleted in favor of mentioning this parameterization in the introduction.

> Typo/style errors:
>
> Line 67: set scalar > set of scalar
>
> Line 92: could be made linear combinations ?
>
> Line 129: by set of ?
>
> Line 294: amio –> amino

These were all changed to improve correctness and readability.

> Line 325: that size are computationally

This was not changed as we were unable to identify the problem with the style or grammar.